# On the Inheritance of Microbiome-Deficiency: Paediatric Functional Gastrointestinal Disorders, the Immune System and the Gut–Brain Axis

David Smith [1,*], Sohan Jheeta [1], Georgina I. López-Cortés [1,2], Bernadette Street [3], Hannya V. Fuentes [1,4] and Miryam Palacios-Pérez [1,4,*]

[1] Network of Researchers on the Chemical Emergence of Life (NoRCEL), Leeds LS7 3RB, UK; sohan@sohanjheeta.com (S.J.); gina.lopez@comunidad.unam.mx (G.I.L.-C.); hanny.v.fuentes@gmail.com (H.V.F.)
[2] Facultad de Química, Universidad Nacional Autónoma de México (UNAM), México City 04510, Mexico
[3] Street Therapy, Kitchener, ON N2H 5C5, Canada; bernadette@streettherapy.ca
[4] Theoretical Biology Group, Instituto de Investigaciones Biomédicas, Universidad Nacional Autónoma de México (UNAM), México City 04510, Mexico
[*] Correspondence: dave.smithathome@gmail.com (D.S.); mir.pape@iibiomedicas.unam.mx (M.P.-P.)

**Abstract:** Like the majority of non-communicable diseases that have recently gained attention, functional gastrointestinal (GI) disorders (FGID) in both children and adults are caused by a variety of medical conditions. In general, while it is often thought that common conditions such as obesity may cause other problems, for example, asthma or mental health issues, more consideration needs to be given to the possibility that they could both be brought on by a single underlying problem. Based on the variations in non-communicable disease, in recent years, our group has been revisiting the exact role of the intestinal microbiome within the Vertebrata. While the metabolic products of the microbiome have a role to play in the adult, our tentative conclusion is that the fully functioning, mutualistic microbiome has a primary role: to transfer antigen information from the mother to the neonate in order to calibrate its immune system, allowing it to survive within the microbial environment into which it will emerge. Granted that the microbiome possesses such a function, logic suggests the need for a robust, flexible, mechanism allowing for the partition of nutrition in the mature animal, thus ensuring the continued existence of both the vertebrate host and microbial guest, even under potentially unfavourable conditions. It is feasible that this partition process acts by altering the rate of peristalsis following communication through the gut–brain axis. The final step of this animal–microbiota symbiosis would then be when key microbes are transferred from the female to her progeny, either live offspring or eggs. According to this scheme, each animal inherits twice, once from its parents' genetic material and once from the mother's microbiome with the aid of the father's seminal microbiome, which helps determine the expression of the parental genes. The key point is that the failure of this latter inheritance in humans leads to the distinctive manifestations of functional FGID disorders including inflammation and gut motility disturbances. Furthermore, it seems likely that the critical microbiome–gut association occurs in the first few hours of independent life, in a process that we term handshaking. Note that even if obvious disease in childhood is avoided, the underlying disorders may intrude later in youth or adulthood with immune system disruption coexisting with gut–brain axis issues such as excessive weight gain and poor mental health. In principle, investigating and perhaps supplementing the maternal microbiota provide clinicians with an unprecedented opportunity to intervene in long-term disease processes, even before the child is born.

**Keywords:** microbiome; non-communicable disease; maternal inheritance; handshaking; immune system; microbial sentinel cells





## 1. Functional Gastrointestinal Disorders: Non-Communicable Disease

Functional gastrointestinal disorders (FGID) are a subset of non-communicable diseases, constituting a wide range of apparently random, seemingly unprovoked disturbances of the gut. They include disorders of hypersensitivity, gut motility, and altered immune function. The development of the field has been overseen by a not-for-profit group called the Rome Foundation, which has issued information and guidance at intervals. The most recent, Rome IV, was published in 2016 and included information on the microenvironment of the gut, the microbiome (albeit mostly the bacteriome), and the gut–brain axis [1]. Interestingly, the connection with the gut–brain axis could perhaps help to explain the superficial similarity of the complex Rome IV deliberations with those pertaining to the brain, the equally complex Diagnostic and Statistical Manual of Mental Disorders, currently in its 5th Edition (DSM-5) [2]. Paediatric FGID represents a subset of such diseases with their own challenges of diagnosis and treatment [3].

The disease term "non-communicable" represents a set of conditions that have recently been increasing in extent and variety but does not seem to be directly related to any infectious agent. While commonly encountered in developed countries, they are an increasing problem in the developing world, albeit starting from a low base [4]. However, what confuses the issue is when seemingly similar conditions arise following an infection, for example, the so-called "long COVID" [5]. Indeed, it may be possible that many immune-related conditions have a hidden trigger such as the potential role of the Epstein–Barr virus in multiple sclerosis [6]. Another feasible cause of non-communicable disease is dietary deficiency, which is especially problematic in otherwise undiagnosed coeliac disease [7] or in dietary intolerance and allergy [8]. Interestingly, the latest idea to explain the growth of microbiome-related disease relies on a combination of dietary deficiency, potentially toxic additives, and high energy, the so-called "ultra-processed foods" [9]. As implied by the title of this article, however, our view of non-communicable disease is as a form of deficiency disease, specifically a lack of function of our mutualistic partner, the intestinal microbiome [10]. In this view, while there is undoubtably a role for a poor diet in non-communicable health problems, it is as an accelerant of disease, rather than as a cause in its own right. Unfortunately, in the absence of a definitive diagnosis, the full extent of such microbiome-deficiency disease is not clear.

Needless to say, many adult weight-related diseases are considered to be some kind of moral failure, and are often blamed on lifestyle factors, albeit incorrectly [11]. As the spread seems to be too fast for classic genetic diseases, this raises the question of maternal microbial inheritance: the microbes transfer from the mother to the neonate during the birth process; additionally, the role of seminal microbiome has more recently been investigated not only as an epigenetic factor [12], but also for the optimal development of the preconception environment [13,14]. Our previous work in this field used the concept of microbiome-function deficiency to draw together the links between obesity and coeliac disease, the latter involving both functional gastrointestinal and immune system disorders and potentially poor mental health [15]. As indicated above, it is important to note that due to the poor absorption of nutrition, sufferers from coeliac disease may present with a BMI (body mass index) in the nominally healthy range [16]. Significantly, a recent publication documented the overlap between the lower functional gastrointestinal disorders, those of the bowel and anorectal system, and coeliac disease, even when the latter has apparently been successfully managed with a gluten-free diet [17]. Interestingly, poor mental health is associated with these conditions and the authors suggest addressing disorders of gut–brain interaction but remain silent on the potential involvement of the microbiome itself [17].

## 2. Investigating the Cause of Disease: Denis Burkitt and Dysbiosis

The first person to document the profound difference between the health of people living a traditional lifestyle and those in the high-income Westernised world was Denis Burkitt. Born in 1911 in Enniskillen, Northern Ireland, UK, Burkitt trained as a surgeon in Edinburgh, UK, and served in Africa during the Second World War. After the end

of the conflict, he went back to the African continent, travelled extensively, and made copious notes about the diet and health of people living their traditional modes of life. He discovered a transmissible viral cancer, now known as Burkitt's lymphoma [18], but was surprised to find a complete absence of the plethora of non-communicable diseases found in "modern Western civilisation" [19]. The diseases identified by Burkitt are listed in Table 1, and are composed of conditions associated with a malfunctioning immune system including cancer and assorted disorders of blood circulation, weight gain, and cholesterol deposition. A product of his time, Burkitt unfortunately did not report any information on mental health issues, but he did report a significant observation, that the faecal output of people in "Westernised" societies was only about one third that of people living in traditional societies, who remained free from non-communicable diseases. The relative retention of energy in people exhibiting low faecal output is the most succinct explanation for weight gain in modern societies [20].

**Table 1.** Selected diseases characteristic of modern Western civilization [19].

| Appendicitis | Coeliac Disease * | Coronary Heart Disease |
|---|---|---|
| Deep vein thrombosis | Diabetes, type 2 | Diverticular disease |
| Gall stones | Haemorrhoids | Hiatus hernia |
| Multiple Sclerosis * | Obesity | Pernicious anaemia * |
| Pulmonary embolism | Rheumatoid arthritis * | Thyrotoxicosis * |
| Tumours of the bowel * | Ulcerative colitis * | Varicose veins |

* Diseases having a significant immune system component.

Observing that the onset of these diseases followed exposure to something in modern life, Burkitt suggested an environmental cause rather than an infectious agent. Lacking knowledge of the microbiota–gut–brain axis, he identified the low fibre intake of modern diets as leading to poor cholesterol metabolism and reduced faecal output, thus giving rise to the bulk of the other conditions listed. However, he admitted that he could not account for the observed immune system deficits simply on the basis of inadequate dietary fibre [19]. As Burkitt himself noted, however, as the cattle herding, steppe dwelling Maasai (Masai in his day) enjoyed a dietary regime similar to that of Westernised peoples, and did not suffer from non-communicable diseases [19], it could be that Burkitt's "Westernised" diseases are not due to dietary factors at all.

The rapid spread of obesity and related diseases nevertheless raised the possibility of some kind of infection. Accordingly, the discovery that the gut microbes of genetically obese mice could transfer enhanced metabolic potential to germ-free mice briefly raised interest [21]. In contrast, around this time, the term dysbiosis was being used to imply a malfunction of the microbiome due to the absence of specific bacteria, the idea being to replace these entities with so-called probiotics. However, it has been pointed out that these ideas have not, as yet, been scientifically validated [22]. Our recent publications have emphasised the lack of microbiome functionality consequently upon reduced microbial diversity as opposed to a lack of specific bacteria, and have therefore sought to introduce heavy metal pollution as a Burkitt-like environmental factor affecting the microbiome. Accordingly, obesity is a consequence of a failure of microbial growth and consequent excretion [15,20].

While the microbiota–gut–brain axis has been related to the onset of obesity [23], it has also been linked to mental health [24]. Our own view is that the microbiome evolved as an "intergenerational" component of the vertebrate immune system in order to calibrate the neonate to recognise the microbial environment of its mother [25]. The main danger of an uncalibrated immune system is the misinterpretation of harmless antigens as representing harmful substances, leading to an epidemic of immune-related disease. Conversely, the immune system may also fail to recognise precancerous behaviour, thereby accounting for the observed rise in cancers apparently resulting from a wide

variety of causes [26]. In this view, the function of the microbiota–gut–brain axis is to supply the microbiome with nutrition by controlling gut motility, while at the same time retaining sufficient nutrition for the survival of the host. The link with obesity follows from the breakdown of this axis, while mental health problems arise from the lack of interoception, the connection between body and brain [27]. In our view, therefore, the appearance of paediatric functional gastrointestinal disorders follows from the inheritance of a malfunctioning maternal microbiome, in combination with the genetic inheritance of the child. While there are elements of diagnosis in this article, its fundamental purpose is the prevention of disease by the assembly of a "birth probiotic" and its use in the repair of the human microbiome (Section 15).

## 3. Summary of Microbiome-Related Concepts and Terminology

In this article, we used the term *microbiota* to indicate the microbial entities themselves, while the term *microbiome* included the genetic abilities relevant to the host organism. It is important to note that a multicellular entity may coexist with several seemingly independent microbial communities, but it is possible that one is preferred, for example, the reported dominance of the gut microbiome within the gut–brain–skin axis that has been associated with acne vulgaris-induced depression [28]. Similarly, the terms *holobiont* and *hologenome* refer to all entities and/or their genes acting together to give each other survival advantages across multiple generations. It is generally considered that the many forms of non-communicable disease is largely a consequence of the reduced diversity of the intestinal microbiome seen across the developed world [29].

As further described below, communication between the two separate entities of the multicellular host and microbial community is by hormone-like *semiochemicals*, molecules passing a message between different species. The microbiome concept makes most sense from an evolutionary perspective, where a microbial community drives the immune status of the multicellular entity [25], possibly via eukaryotic *microbial sentinel cells*, potential precursors of the antigen-presenting dendritic cells of the immune system [30]. In this context, we note that the immune system is involved with the gut–brain axis, in what we have termed an *immune/semiochemical system* [31]. Finally, we used the computer science term *handshaking* to describe the initial interaction between the genetics of the neonate host and its newly inherited "guest" microbiome, possibly even within the first few hours [27]. Significantly, the concept of neonate-handshaking accounts for the difficulty experienced in attempts to ameliorate non-communicable diseases such as allergy, for example, by increasing the microbial diversity in either adults or children [32].

## 4. Evolution: Lynn Margulis and Carl Woese; A Vertebrate Holobiont

At about the same time that Denis Burkitt was describing his African experiences, Lynn Margulis, writing as Lynn Sagan, was describing her ground-breaking work on the symbiotic origin of eukaryotes [33]. Taking the concept further, she later introduced the "symbiosis in evolution" concept as the holobiont, the idea that a species should be treated as a combination of every living entity involved in its biology [34]. A logical development of this theme is the hologenome, in which the critical component is the expression of genes providing functions of some value to the host [35]. While the bulk of symbiosis work is being undertaken on commercially sensitive invertebrates [36], it could be that such animals interact too easily with their environment, introducing unhelpful complexity. As an example of such flexibility, the ability of invertebrates to gain potentially valuable genes by horizontal gene transfer (HGT) continues to surprise. A recent example is the extensive uptake of plant genes into the genome of the whitefly herbivore *Bemisi tabaci* [37].

Paradoxically, it may be that the microbiome of vertebrate animals will prove to afford a clearer example of symbiosis than the nominally less complicated invertebrates. While Darwin espoused the idea of the survival of the fittest, this idea has dominated the discussion of genetic evolution for decades, with symbiosis and other forms of change largely relegated to an afterthought. Instead, it seems that "evolution" on the unicellular

level occurs largely by HGT, a concept championed by Carl Woese using the expression "Darwinian threshold" to distinguish the difference between the two forms of evolutionary change [38]. In recent articles, our argument has been that the animal-microbe holobiont gains the best of both sides of Woese's threshold, the stability of the multicellular host on one hand, allied to the flexibility of its diverse microbial guest on the other [25].

Logic suggests that the basic function of a symbiotic microbiome guest is to support the immune system of its multicellular host while obtaining nutrition in return [10]. In this context, while the development of the first vertebrate animals across the Ediacaran–Cambrian boundary may have contributed to their efficiency as predators [39], the elongation of their body plan would have posed problems for the heritability of key microbes from within their microbiomes. Accordingly, it seems that the outcome of this challenge was a division of function: while the major cognitive apparatus lies at the front of the animal, under the control of the brain, the primary microbiome resides at the end of the digestive tract, allowing relevant microbial constituents to be passed on to the next generation. Accordingly, the most succinct explanation for the immune system component of non-communicable disease is that a form of microbial sentinel cell exists within the properly functioning microbiome, and that this class of cell has been lost under the conditions of modern life [25]. While the microbiome has often been treated as an ad hoc collection of environmental bacteria introduced via our food, more recent evidence has traced an origin in humans back to our African origins [40]. Of course, a sentinel cell will be more complex than simple prokaryotes, and although unicellular eukaryotes are known to exist inside the microbiome, their function is not completely characterised. One example of such entities includes members of the genus *Blastocystis*, classified as parasites, but known to be associated with normal weight, apparently healthy individuals [41]. The fungal "mycobiome" has also been studied, especially within early childhood [42], and a suggestion has been made that microbial eukaryotes could be the missing link in gut microbiome studies [43]. Note that we have dealt with these concepts in greater detail in a recent publication covering both the evolutionary aspects of vertebrate animal–microbe symbiosis as well as its loss in the polluted environment of high-income countries [25].

## 5. Epidemiology: David Strachan and David Barker; An Infant Origins Hypothesis

In September 1989, David Strachan published the results of his observations concerning the spectacular rise of diseases such as asthma and seasonal allergic rhinitis: "hay fever, hygiene, and household size" [44]. In essence, what became known as his "Hygiene Hypothesis" suggested that atopic and autoimmune diseases start in children that live in modern environments, the key point being that the absence of microbes is capable of training their immune systems. This idea was taken up by Graham Rook and his team as the "Old Friends" concept of external training agents. In spite of intensive efforts, by 2013, it was clear that such an external agent could not be found [45]. Nevertheless, levels of atopic disease kept increasing, with seemingly new conditions such as food allergy raising concern [46]. The most important feature about these immune system disorders is that they originated in the young, typically being observed in a characteristic sequence described as the "atopic march" [47]. In addition, it also became clear that animals associated with humans suffer from a very similar series of immune system problems, suggesting a common mechanism, at least across the Mammalia [48].

At almost the same time as Strachan was writing, in November 1990, David Barker published his own article "The foetal and infant origins of adult disease" [49]. Using the subtitle "The womb may be more important than the home", his idea later became known as "The Foetal Origins Hypothesis". Although he was mostly describing weight gain leading to stroke and heart disease, it is important to note that he specifically mentioned schizophrenia in his original article [49]. Although successful when studied from an economic perspective [50], its fundamental problem is the absence of an acceptable underlying mechanism, as confirmed by a later review [51]. Whatever the fundamental cause, obesity and related issues are increasing, as illustrated by a study of sport-related

statistics measured in different batches of 10-year-old schoolchildren in 1998, 2008, and 2014. Although the overall BMI did not change, the height and weight both increased over this period, and vital attributes such as handgrip and leg strength both declined [52]. Similar results were found in an independent study [53]. It is important to note, however, that Barker's initial concept applies to the birth process in general and could therefore be termed as an "Infant Origins Hypothesis", laying the emphasis on the second part of the title of his original paper [49].

### 6. A Microbiome-Health Hypothesis: "Handshaking" and the Birth Process

Interestingly, there are a variety of conceptual overlaps between biology and engineering. In particular, "computer viruses" possess characteristics similar to biological viruses. Perhaps analogously, as described in the Concepts and Terminology section, while the engineering term "handshaking" refers to the exchange of protocols of communication between two independent electronic devices, it may be that the same term could be employed to indicate interactions between the microbiome and the gut wall of the newborn animal. In its current format, our Microbiome-Health Hypothesis suggests that this "handshaking" requires a fully functional microbiome to be in place immediately after birth (or after hatching from an egg), in order to allow for the proper development of the gut–brain axis. Of course, in humans, there are many years between neonate and adult, and it is likely that this axis will need a supplement to the handshaking process after key events such as puberty. There is little doubt, however, that the early days are the most important, quite possibly the first hours. It seems clear that the period immediately after maternal microbial inheritance is of great significance to the health of both the child and adult [54,55]. While it is likely that the microbiome has its effect through epigenetic mechanisms [56], it is clear that much remains to be done [57].

Modern antiseptic methods ensure that babies can be safely delivered by caesarean section (C-section) under a sterile environment. Interestingly, the same procedures can be used for the obtention of gnotobiotic, so-called germ-free, laboratory animals [58]. It is known that laboratory mice have modified physiological responses to stress under these conditions, but that post-natal reintroduction of gut microbes eliminates many of their problems [59]. Altered gut microbes have a similar effect in humans, as a later study involving children experiencing problems with caregiving has shown [60]. The relationship between C-section delivery and obesity has been studied most often, usually finding a positive correlation. For example, a study by Yuan et al. showed a 15% greater chance of the development of obesity after C-section birth, with "additional research needed to clarify the mechanism" [61]. When a swabbing procedure called "vaginal inoculation" took place, Dominguez-Bello and her team demonstrated that the microbiota of those children showed some similarity to their vaginally delivered peers. Health studies are now underway [62]. A similar study by Chu et al. compared the maturation rate of the microbiome between C-section or vaginal methods of delivery, showing that there was no apparent difference six weeks post-natal [63]. Likewise, the role of birth mode in the possible transmission of obesity from mother to child was followed by bacterial 16S rRNA, suggesting that species from the phylum Firmicutes were most likely to be responsible [64], but they did not check for the presence of unicellular eukaryotes. In contrast, a similar study following the transfer of vaginal fungi in addition to bacteria identified lower mycobiome diversity as contributing to type 1 diabetes in children [65].

Of course, this vaginal inoculation procedure can only transfer the maternal microbiota that are already present and is therefore of no use to the baby if such microbiota were not functioning properly in the mother. In addition, while simpler cells such as bacteria can be picked up from the environment more-or-less readily, unicellular eukaryotic sentinel cells are more likely to be specific to the individual and would not be expected to be recovered once degraded. In this fashion, microbiome quality can only worsen across the generations, an observation that is consistent with the epidemiology of non-communicable disease [66]. However, the C-section procedure is convenient and often necessary, and is

increasing worldwide [67]. Furthermore, the incidence of C-section delivery is predicted to significantly increase in the future [68]. Finally, regardless of how "natural" the concept of vaginal inoculation/seeding, there remains a reluctance to carry out a procedure that has neither been subjected to stringent clinical trials nor has a robust "theory of microbiome action" to guide its practitioners [69].

At this point, it is worth bearing in mind the comments of Brüssow on the continuing need to confirm the relevance of suspect microbes by reference to the principles exemplified by Koch's postulates [22].

*6.1. Natural Birth: Efficient Microbial Transfer*

Three components common to all the Vertebrata include the brain (acting with its body), the intestinal microbiome, and a connecting gut–brain axis (Figure 1, left hand side). As semiochemicals act on the gut wall, the gut–brain axis induces peristalsis, passing nutrition down to the microbiome. Along with the nutrition, environmental antigens prime the hypothetical microbial sentinel cells ready for the next stage, as the animal creates its offspring. The centre section indicates the natural birth process, where the neonate becomes an independent entity, either via live birth or egg. It is at this stage that direct "contamination" takes place, transferring the microbiome to the infant [70]. Interestingly, bacteria are present in many of the structures associated with birth, but it is not known as to what degree they contribute to either the success of the pregnancy or the health of the baby [71]. The birth process is completed as illustrated on the right-hand side, as semiochemical production initiates "handshaking", and microbial sentinel cells calibrate the immune system relative to the microbial environment of the mother. At this stage, mammals have evolved an extra layer of assistance, as mother's milk provides nutrients [72] and possibly supplementary microbes [73].

*6.2. Delivery by Sterile Caesarean Section: Limited Microbiota Transfer*

The situation described in Figure 2 represents a healthy adult, with a fully functioning microbiome and gut–brain axis (left hand side), being delivered of her baby under sterile hospital settings (centre section). Nevertheless, hospital "sterility" is not as severe as for the birth of germ-free animals, and slow contamination will occur under these conditions. The key point here is that, as stated above, the microbiome of the child will eventually establish itself, according to the findings of Chu et al., but, of course, it may be too late for a completely effective handshaking function or immune system support [63]. Disease results from either the absence of an effective gut–brain axis such as poor mental health or inadequate "calibration" of the immune system, leading to problems such as autoimmune disease (right hand side). A final point is that any microbes that do not manage to spread to the infant will not be present in future births, affording what can be described as a "snowball effect" of increasing problems across generations.

Of course, the birth of twins poses special dangers for the mother, and planned C-section deliveries are often carried out. Unless the vaginal inoculation procedure [62] is carried out, each twin will have different levels of contamination, and hence a different chance of disease. While twin studies (genetically identical versus fraternal twins) have been used to study the vexed question of "genes versus environment"; only a few decades ago, delivery by C-section was not normally considered, implying that similar studies carried out today would yield different results. We have reported our conclusions more fully elsewhere [27].

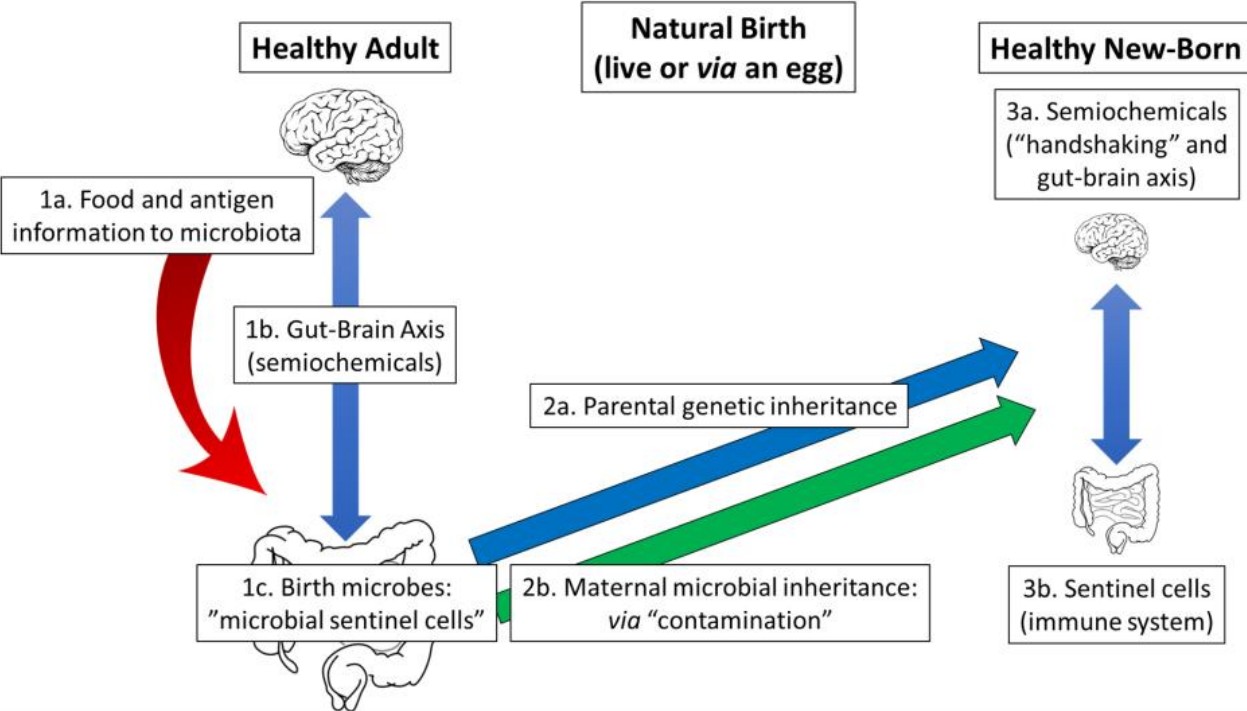

**Figure 1.** Natural birth. *Left hand side*: Indicates the function of the gut–brain axis in the healthy vertebrate adult. In essence, this axis acts to partition nutrition between the body (the host) and the microbiome (the guest). Box 1a. Alongside food, antigen information regarding the external microbial environment is passed down to the microbiome for subsequent transfer to the next generation, with microbial sentinel cells acting as an "intergenerational" component of the immune system. Box 1b. The microbiome produces semiochemicals, hormone-like molecules that pass information between different species, in this case, "interkingdom", from prokaryotes to eukaryotes, acting to stimulate the gut wall, and therefore the gut–brain axis. Finally, Box 1c represents hypothetical microbial sentinel cells ready to be transferred to the neonate by apparently accidental contamination. *Centre*: This represents the vertebrate birth process that has evolved since the Precambrian. Box 2a. The foetus develops according to the genetic inheritance of the parents and Box 2b is contaminated by the microbes from the intestine of the mother on birth: the "maternal microbial inheritance". *Right hand side*: Illustrates the newborn animal, with complete parental genetic and maternal microbial inheritances. Box 3a. Semiochemicals produced by the microbiota in the intestine interact with the gut wall in a "handshaking" process, allowing the growth of the gut–brain axis. Box 3b. The sentinel cells transferred from the maternal microbiome interact with the gut wall in their own version of the handshaking process. Their output is part of the combined immune/semiochemical system interacting with the gut–brain axis [31].

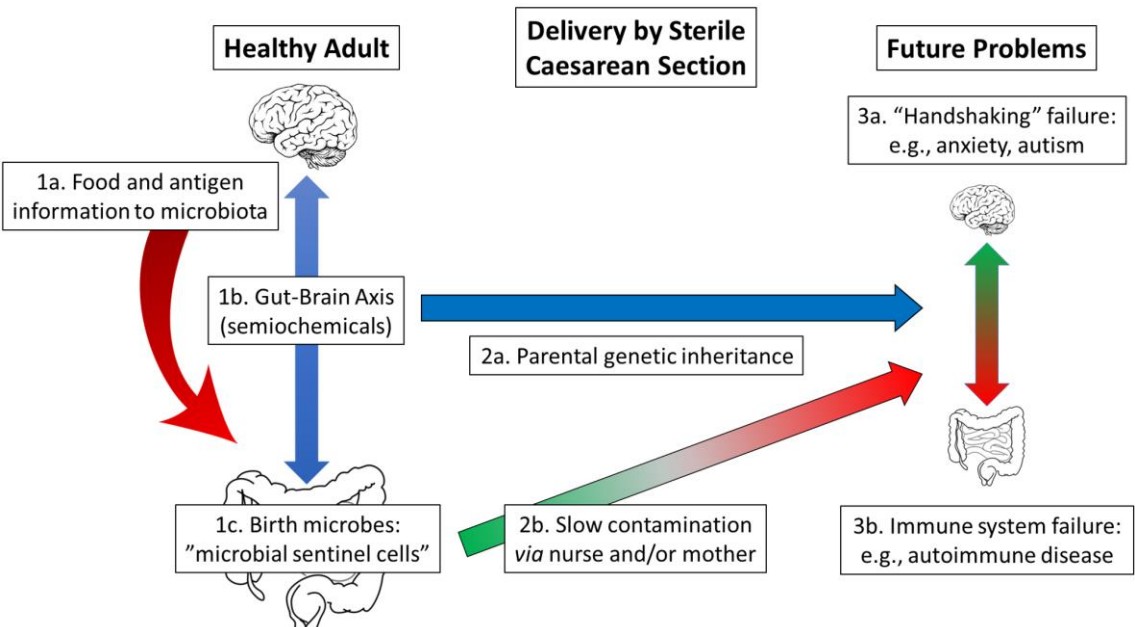

**Figure 2.** Delivery by caesarean section under hospital conditions. *Left hand side*: Represents a healthy vertebrate adult with, Box 1a, the transfer of food and antigen information to the microbiome, Box 1b, semiochemicals produced by the intestinal microbiota, and Box 1c, "birth microbes" ready to be delivered to the baby. *Centre*: Indicates that the two halves of the natural birth process have become separated. Box 2a represents the parental genetic inheritance of the baby while Box 2b illustrates the difficulty experienced by the maternal microbiota to reach the baby. *Right hand side*: Represents the chance of disease as the child grows into an adult. Box 3a describes the effect of the loss of interoception due to the absence of a fully functioning gut–brain axis on the developing brain, with the potential for conditions such as anxiety and autism. Box 3b indicates problems with the immune system, potentially autoimmune disease, as it is no longer calibrated by the presence of microbial sentinel cells mainly from the mother.

## 7. A Microbiome-Health Hypothesis: Peristaltic Control and Antigen Recognition

As described in the previous section, the essence of symbiosis is for both components to reap the benefits of collaboration. Whether through microbial sentinel cells or by some other mechanism, the first point is that the host benefits from the immune system impetus provided by the microbiome. In return, the microbiome receives a share of the nutrition of the host in a carefully controlled process that has evolved to keep both components viable through the generations, even under adverse circumstances such as famine or illness. The most succinct rationalisation of this process is if the rate of peristalsis is mediated by the gut–brain axis, as are the all-important actions associated with pregnancy and birth [31]. As described in the Concepts and Terminology section, the microbiome exerts its actions through the effect of semiochemicals, substances produced by one species in order to affect the behaviour of a different species. An example of such a semiochemical is the scent produced by a flower to attract a pollinating insect, while conversely, communication within the gut lumen requires hormone-like water-soluble substances. Interestingly, it has been suggested that the genes required to synthesise the water-soluble signalling molecules found in the body could have been passed on from bacteria by horizontal gene transfer at intervals during evolution [74]. Furthermore, it seems most likely that such molecules exert their effect directly on the gut wall, thereby activating the gut–brain axis. Interestingly, alongside previously known enteroendocrine chemosensors [75], a new class of so-called "neuropod" cells, gut sensory epithelial cells forming synapses, have recently been characterised that transduce sensory signals directly from the gut contents to the vagal nerve and hence the brain [76]. It may be that such cells respond to biogenic amines such as dopamine, which are produced by microbial action within the gut lumen [77]. Indeed,

this may be the fundamental mechanism in which the status of the microbiome affects both the mental state and physiological responses.

Although nutrition is provided to the intestinal microbiota, antigens representing the local microbes are also passed down to the microbiome. At this point, our suggestion is that the hypothetical sentinel cells, referred to above, will be able to distinguish potentially harmful microbe-derived antigens from the harmless proteins found within food or on the surface of pollen grains. In this hypothesis, such information will be passed on from the female to the next generation in order to calibrate the naïve immune system of their offspring against her microbial environment. It is important to note, however, that the immune system of the adult animal is also supported by the microbiome. The most obvious way in which this occurs is through the production of short chain fatty acid (SCFA) salts including acetate, propionate, and butyrate [78], probably more as a by-product of metabolism under anaerobic conditions rather than manufactured for a specific reason. Nevertheless, these substances act to power adjacent cells, both serotonin for the gut–brain axis [79], and to regulate aspects of the immune system [80]. As this system is closely involved with non-communicable disease, we refer to the gut–brain axis as an immune/semiochemical system [31]. In addition, we also suggest that stimulation of the immune system by probiotics may have disease-ameliorating effects via the gut–brain axis, perhaps accounting for the placebo effect, in part at least [27].

## 8. Current Understanding of the Neonate Immune System

The research on the immune system of both the foetus and neonate is ongoing, and has been reviewed elsewhere [81,82]. In general, the immune system of the neonate is suppressed by the secretion of anti-inflammatory cytokines, poor lymph node architecture, and reduced expression of cell activation markers (e.g., HLA-DR, CD80 in dendritic cells; CD107c, CD54/ICAM-1 in natural killer cells, etc.), nevertheless, innate lymphoid cell functionality is increased and regulates microbiome colonization [82,83]. Indeed, microbiome colonization and epigenetic modifications are the principal factors that influence neonatal innate immunity [81]. The neonate has to adapt and learn to respond as it changes from the nearly sterile and immunosuppressive environment of the womb to the natural world, full of new stimuli. In order to protect the infant while maturing its adaptive immune system, the mammalian mother passively transmits antibodies through her breastmilk, along with plenty of nutrients and molecules beneficial for the colonizing microbiota. Additionally, breast feeding is crucial for generating a tolerogenic environment and priming the immune cells, since human milk oligosaccharides have immunoregulatory properties [84].

Evidence of how nutrition during pregnancy could affect the child's immune system is vast. In a mouse model, the supplementation with galacto-oligosaccharides/inulin during gestation increased the frequency of $B_{reg}$ cells both in the mothers' uterus and placenta, and in the foetal bone marrow and intestine [85]. Furthermore, after 7 weeks after birth, an increase in the number of $CD9^+$ $B_{reg}$ cells at the mesenteric lymph nodes was detected, indicating that some nutrients favour the establishment of a tolerogenic immune profile in the foetus. Additionally, the ingestion of fructo-oligosaccharides during gestation increases colostrum, and milk IgM antibodies [86] protect the baby while an effective microbiome develops. The mechanism of how these oligosaccharides enhance the production of IgM isotype remains unknown, but it is noteworthy that in modern-day populations, fructo-oligosaccharides are fermented by the mother's *Lactobacillus* and *Bifidobacterium*, and milk glycoproteins increase the colonisation of these same bacteria in the newborn by providing oligosaccharides [87].

Some of the dietary components and microbiota-derived metabolites can alter the balance of immune homeostasis and other systems [88]. For example, oligosaccharides and dietary fibre are fermented by *Clostridium*, *Faecalibacterium*, and *Roseburia*, among others producing SCFAs (saturated hydrocarbon chains composed of one to five carbons bound to a carboxyl group), of which the most studied are acetate, propionate, or butyrate; some of their receptors are G protein-coupled receptors (e.g., GPR41, GPR43, or GPR109A) [89].

Interestingly, the SCFA propionate promotes the differentiation of naïve CD4$^+$ T cells to T$_{reg}$ while long-chain fatty acids in the small intestine promote differentiation towards T$_{h1}$ and Th$_{17}$ profile [90]. Furthermore, it has been reported that both propionate and butyrate enhance dendritic cell TGF-β production, which induces T$_{reg}$ differentiation in vitro [91]. Smith et al. showed that SCFA supplementation to germ-free mice increased T$_{reg}$ cells but neither T$_{h1}$ nor T$_{h17}$ in the colon [80]. Note, however, that germ free mice have limited microbiomes and potentially a dysregulated immune/semiochemical system. T$_{reg}$s are necessary to suppress the local immune response by expressing the transcription factor Foxp3, which induces the secretion of IL-10 and TGF-β, however, the balance with inflammatory responses is crucial for defeating against pathogens (i.e., by T$_{h17}$). The balance between T$_{reg}$ and T$_{h17}$ cells influence intestinal homeostasis. The shift towards any side of the balance is associated with autoimmunity, cancer, and metabolic diseases. While T$_{reg}$s are a subset of T cells that suppress the immune response, T$_{h17}$ cells coordinate the immune activation towards microorganisms. Although they have opposing functions, they both share regulating mechanisms [92].

In the large intestine, tryptophan is transformed into different bioactive indole derivatives by different bacteria such as *Escherichia coli*, *Micrococcus aerogenes*, *Paracolobactrum coliforme*, *Proteus vulgaris*, and different species of the genera *Clostridium* and *Lactobacillus* [86]. Indole-derived metabolites regulate the immune response and mucosal integrity via binding to aryl hydrocarbon receptor (AhR) in the cytoplasm [93]. In turn, AhR plays an important role in intestinal homeostasis, maintains mucosa integrity, and is associated with chronic inflammatory diseases in the intestine and other tissues. The activation of AhR through indole metabolites induces the differentiation of T$_{reg}$ cells and supresses T$_{h17}$ activation. The polyamines are molecules found in food as well as a metabolite product of some bacteria. Spermidine, one of the most studied, alters the T$_{h17}$/T$_{reg}$ ratio and induces macrophage differentiation towards the M2 profile by the upregulation of arginase-1, which is essential to perform anti-inflammatory activities. Additionally, spermidine reduces NF-κB activation, and therefore inflammatory genes [94]. Just like these, microbiome metabolites could directly alter immune and other systems.

To study how the gut is first colonized, it is important to discover which microbes and factors are transmitted prior to the "handshaking" process during, and possibly before, delivery by natural birth. Indeed, many of these factors will be introduced via the skin of the newborn, whether human or animal, and this gives us an opportunity to discover their nature (see Sections 11–13). In accordance with our hypothesis, these pioneers should settle as the first microbial populations and have the first interactions to prime the immune system of the newborn. Note that the immune system of the foetus is functional and capable of responding to danger signals [83]. Nevertheless, it still has to mature and develop specific T and B lymphocytes [81]. The immune system in the gut contributes to generating a suitable niche for the first colonizers to settle. Additionally, bacteria from the genera *Bacteroidetes*, *Firmicutes*, *Proteobacteria*, *Actinobacteria*, *Enterobacteria*, *Bifidobacterium*, and *Akkermansiaceae* generate an anaerobic environment and help to regulate the pH and metabolites needed by other microorganisms. In a similar fashion, the study of microbial sentinel cells could open an avenue in the microbiome field with potential use for establishing a healthy gut. As for factors transmitted during "handshaking", sentinel cells may regulate the immune response and switch towards a tolerogenic profile.

In this context, it is important to note that the above results were obtained by studying modern-day populations and although individual subjects may seem to be healthy, they are likely to have the potential for the future development of non-communicable disease [66]. In particular, it is worth noting that atopic diseases such as babyhood eczema have been known to develop within the first month [47], well within the timeline of many of the above quoted studies. In addition, due to the readily accessible ribosomal 16S subunit, the great majority of microbiome studies actually refer to the prokaryotes, rather than to any microeukaryotic component, which may represent a sentinel cell. Indeed, it is possible that the bacteria referred to above are actually only dominant as a consequence of

modern-day pollution, rather than being of significance in their own right. An early report on the microbiome of the Hadza, a Tanzanian group only marginally exposed to modern pollutants, did not, for example, mention bifidobacteria [95]. In our view, the value of a microbiome lies in its function, while dysbiosis is a microbiome-function deficiency disease.

## 9. The Nature of the Microbiome: Mobile Genetic Elements

While pneumococcal disease is caused by the bacterium *Streptococcus pneumoniae*, not all of its various strains are dangerous to life. In 1928, a researcher named Frederick Griffith published some observations on the transfer of lethality between two such strains of *S. pneumoniae*. In essence, he killed one of the dangerous strains and showed that a "transforming principle", now known as a mobile genetic element, enabled the transfer of its lethal ability to a previously harmless strain [96]. While this experiment was later used to confirm that DNA-based genes, rather than proteins, were the principle of heredity [97], his observation makes the important point that the gene-expressed function of a suitably diverse microbiome, the way in which it coordinates with the body, may be independent of its individual microbial constituents. At the most fundamental level, the maternal microbial inheritance of a child presumably acts through epigenetic modification of its parent-derived genes [56], ultimately calibrating its response to the microbial environment of the population into which it is born [10]. It is interesting to note that while pathogenicity is not likely to be an evolved attribute, the mutualistic interaction between the host and microbiome ultimately ensures the survival of both components and, indeed, of the holobiont itself [34].

An observation regarding nutrient cycling in the upper layers of the sea led to the discovery of the effect of bacteriophage viruses upon microbial populations in closed systems. It seems that the fragments remaining after phage-induced bacterial lysis were reassembled into new bacterial growth more quickly than they could be removed from the system. This effect was termed as the "viral shunt", as phage action led to a net change in the microbial population distribution rather than a reduction in the overall microbial population density. In other words, the bacteriophage action increases the rate of change of the microbial composition [98]. This same effect has also been described in soil compartments as "biogeochemical turnover" [99]. This process allows for the mobile genetic element-derived exchange of potentially useful functionality such as the ability to catabolise red seaweed glycans taken up from external microbes [100] (see our earlier work on the origins of the microbiome [25]).

## 10. The Degradation of the Microbiome: The Role of Heavy Metal Pollution

Although the loss of potentially critical species following the delivery of babies by caesarean section under sterile conditions is the most obvious reason for the observed reduction in microbial diversity [29], it does not fit with Burkitt's observation of high levels of non-communicable disease within a few decades of the end of the Second World War, when delivery by sterile C-section was rare [19]. Three alternatives have been mentioned.

### 10.1. Diet-Induced Extinctions

Denis Burkitt was an effective communicator, and made his view very clear: that the diseases of "modern Western civilisation" were due to the lack of dietary fibre. In spite of the fact that he contradicted himself in the case of the Maasai [19], by and large, researchers have accepted this idea. A particularly persuasive piece of research was carried out by Sonnenburg and his team in which populations of mice were only allowed access to food containing low levels of dietary fibre. Over several generations, it was observed that the diversity of their intestinal bacteria was reduced [101]. When this observation was tied in with the importance of short chain fatty acids, Burkitt's ideas were revisited [102]. In our opinion, however, his Maasai observations refute these hypotheses [15,20]. Indeed, on the whole, it seems more likely that a low diversity microbiome causes obesity following lowered gut motility, as described above. We have described our view of the relationship between food quality and the microbiome in a recent publication [31].

### 10.2. Antibiotics

Although there has been an increase in the level of antibiotic use since the end of the Second World War [103], it seems to have had little effect on the gut microbiome, at least as far as the bacteria are concerned [104]. Instead, it seems that one role for the caecal appendix is to act as a reservoir for microbes, and this may prevent the total loss of valuable entities under the influence of a course of antibiotics [105]. While antibiotics act to eliminate bacteria in concert with immune and lymphatic systems within the body, according to the viral shunt mechanism (see Section 9), they also act to skew microbial populations within the closed system of the gut lumen. As such, it is feasible that antibiotics may cause a temporary loss of potentially valuable microbiome function, which may affect the handshaking process, especially if given around the time of birth. Growing children may also be vulnerable to the distortion of microbiome function. However, these effects are not limited to people, and they have found extensive use in the enlargement of farmed animals such as swine [106], chickens [107], and, possibly even fish [108]. It is interesting to note that common antibiotics also affect microeukaryotes [109], but there is no reason to suppose that microbial sentinel cells would be eliminated in this way.

### 10.3. Heavy Metal Toxins

Within the context of the viral shunt mechanism, the gut lumen is a hive of activity, with a suitably diverse collection of ever-changing microbes theoretically capable of digesting almost any organic substrate that is commonly found in the natural environment. While plants have evolved methods to safely sequester heavy metal ions, the mobility of animals has rendered such detoxification unnecessary, as highly polluted areas were restricted to the weathering of ore outcrops, and animal evolution would occur away from these areas. Accordingly, one thing that the animal holobiont will be unable to deal with is significant exposure to sufficient concentrations of toxic heavy metal ions to disable the microbiome [25]. Indeed, barring sequestration, the only effective defence would be rapid excretion, but the Microbiome-Health Hypothesis posits that the result of such poisoning is a slowdown in the rate of peristalsis, and hence the rate of excretion [20]. Although the term "constipation" refers to an ill-defined change in bowel habit, it can be associated with heavy metal poisoning [110]. The epidemiology of non-communicable disease is also consistent with exposure to heavy metal salts. The first person to describe hay fever (seasonal allergic rhinitis) was John Bostock in England in 1819 [111]. Intrigued, he later discovered that there were few cases, all of them from the very highest ranks of society including royalty [112]. We assigned this phenomenon to the use of cosmetic heavy metal salts [113] poisoning the microbiome of his mother to just the right degree to affect her son. Another early sign of the onset of non-communicable disease is provided by records relating to United States Army military personnel, which showed a steady decrease in the average body temperature over 150 years since the mid-19th century [114]. We assigned this reduction in body temperature to the well-known phenomenon of energy compensation following the apparent increased levels of exercise as the average body weight is raised from one generation to the next [15]. In summary, it seems that this observation reflects both the beginning of industrial pollution in the United States of America, and perhaps, also the beginning of their obesity crisis. Studies of the effects of heavy metal pollution on the human microbiome are still in their early stages [115], and may be a significant factor affecting insect life [116].

In a more specific episode, tetraethyllead was added to petrol around the world throughout the bulk of the 20th century, starting in the 1920s, at least until around 2000 [117]. In addition to any long-term effects on humans, we suggest that wild animals living in the vicinity of such high-pollution activities could suffer from lead poisoning, either directly or via the microbiome, especially English populations of the European hedgehog, *Erinaceous europaeus* [66].

## 11. FGI Disorders and Non-Communicable Disease: Drawing the Threads Together

While comparing traditional peoples with those of "modern Western civilisation" Burkitt pointed out the tendency for several different types of non-communicable disease to occur in the same individual [19]. This is especially true for functional gastrointestinal disorders and similarly ill-defined conditions such as coeliac disease [17] whereas, in turn, coeliac disease has been associated with poor mental health [118]. Of course, an association between different conditions should not be taken to imply that one causes the other. Indeed, a recent study has used Mendelian randomisation to investigate the nature of links between allergic disease and mental health. Their conclusion was that any such links are unlikely to be causal [119]. Significantly, our Microbiome-Health Hypothesis implies a parallel relationship between immune system and mental health, with both obesity and functional gastrointestinal disorders dependent upon the control of peristalsis via the gut–brain axis. These themes are illustrated in Figure 3.

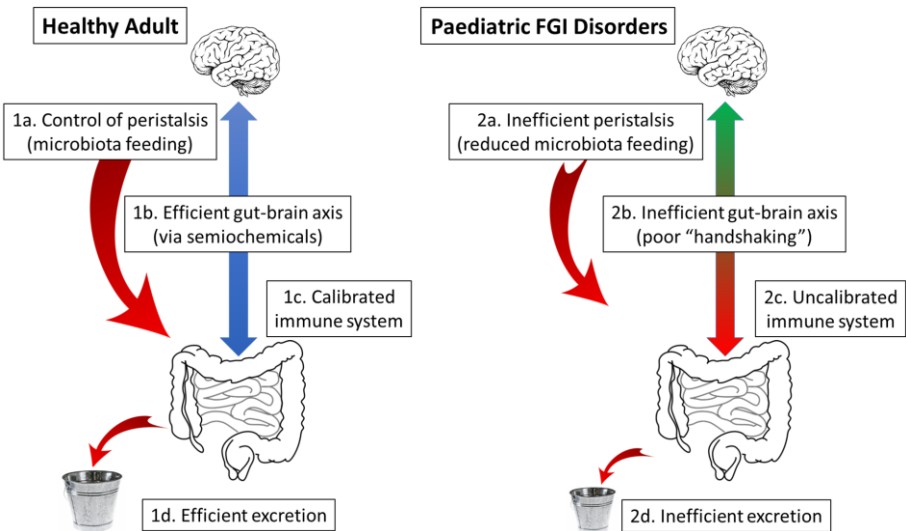

**Figure 3.** The roots of paediatric non-communicable disease. *Left hand side*: In this flow diagram, the healthy adult exhibits: Box 1a, good control over gut motility, Box 1b, an efficient semiochemical-guided gut–brain axis, Box 1c, an effective immune system, and Box 1d, In addition to the absence of non-communicable disease under normal conditions with adequate nutrition, a high faecal volume is to be expected following the observations of Denis Burkitt [19]. *Right hand side*: This diagram represents the presence of non-communicable disease, potentially in all of its forms. Boxes 2a and 2b refer to the consequences of an inefficient handshaking process, with a weak gut–brain axis and lowered gut motility. Box 2c describes an "uncalibrated" immune system as a result of the absence of microbial sentinel cells, while Box 2d represents lowered faecal energy excretion as a consequence of inadequate feeding of the intestinal microbiota. In turn, granted that food energy intake and gross exercise levels remain unchanged, weight increases until a new balance in energy expenditure is reached, as described in an earlier publication [20]. Finally, it is worth emphasising that the difference between the right- and left-hand sides of this figure closely matches Burkitt's description of the diseases of "modern Western civilisation" [19].

### 11.1. The Healthy Animal: A Virtuous Circle

According to this Microbiome-Health Hypothesis, while both the brain and the microbiome vie with one another for food, it is the balance between these two competing influences that determines gut motility, the rate of peristalsis (Figure 3, left hand side). While there is essentially a constant demand for nutrition from the microbiome, the request from the body will depend on circumstances, increasing with famine or illness. In these circumstances, peristalsis becomes inhibited, gut motility decreases, and more food is absorbed into the body. In turn, the microbiome will power down in an organised manner, remaining viable until the emergency passes, and the flow of nutrition may resume

(Boxes 1a and 1b). As unicellular eukaryotic microbial sentinel cells are passed from the mother to neonate, and an adequately fed microbiome can produce SCFAs, the immune system is calibrated against the microbial environment into which it is born (Box 1c). We have previously described this efficient semiochemical-induced self-feeding of the microbiome as a virtuous circle, leading to benefits for both the host and guest, not least because both components are passed on from one generation to the next (Figure 1, above) [10]. Of course, the whole maternal environment has an impact on the formation of the microbiota, especially bacteria, and this is linked to the risk of childhood chronic disorders that may continue into adulthood [120]. Moreover, the consequences of a vicious circle can be passed on for at least four generations, as has been demonstrated in mice [101]. Interestingly, in normal circumstances, with high levels of nutrition to the properly functioning microbiome, microbe growth and efficient peristalsis-induced excretion leads to the high faecal output described by Denis Burkitt (Box 1d) [19].

### 11.2. The Lack of Handshaking: A Potential Cause of Paediatric FGI Disorders

In this hypothesis, the failure of the initial "handshaking" process following inadequate exposure of the neonate to a fully-functioning maternal microbiome (e.g., by caesarean section, Figure 2) has a series of effects, as summarised in Figure 3, (right hand side). While the primary effects are reduced access to microbial sentinel cells and semiochemicals, the consequences are inefficient peristalsis and a weakened gut–brain axis, with an absence of interoception eventually causing poor mental health and an uncalibrated immune system (Boxes 2a–c). The characteristic features of paediatric functional gastrointestinal disorders may be present: reduced gut motility and altered immune function, with hypersensitivity presumably following on from inflammation and poorly coordinated muscle contractions. As mentioned earlier, these symptoms may merge into full-blown coeliac disease [17]. In addition, the reduction in nutrition being delivered to the microbiome eventually results in low faecal growth and potentially obesity [15].

Another example of diseases associated with poor gut motility in the adult is varicose veins, ascribed by Burkitt to faecal compaction [19]. However, as defecation is a socially significant act, the desire to complete the process must be consciously deferred until the appropriate moment when, according to this hypothesis, the signal to initiate peristalsis needs to be sent via the gut–brain axis. Should the gut wall fail to receive this signal, it must be compensated for by straining, producing an intra-abdominal pressure of up to 200 mmHg, enough eventually to render the lower leg valves incompetent. Along with varicose veins, both haemorrhoids and hiatus hernia may result from the same cause, while Burkitt suggests including deep vein thrombosis in this cluster of conditions [19].

### 12. Loss of Microbiome Function: When Is a Holobiont Not a Holobiont?

It can be seen that there are many ways in which microbial diversity can be lost in the context of the modern world and sometimes that can cause confusion. As an example, while studying amyotrophic lateral sclerosis using a mouse model, it was found that two laboratories reported different results. Careful investigation showed that the two teams used mice bred by different suppliers, and that one mouse population lacked a bacterium with an effect on inflammation [121], perhaps casting doubt on which model is the most valuable in the context of human disease. Of course, many studies use laboratory mice as models for human studies, and similar problems will doubtless go unrecorded. As a potential example, excessive anxiety is a serious problem in "Westernised" societies, and a study was undertaken to investigate the effect of a widely-used sweetener, aspartame, on the amygdala of mice. Compared to plain water, aspartame-laced water produced diazepam-neutralised anxiety behaviour that was shown to be due to a shift in the excitation–inhibition equilibrium in the amygdala. Interestingly, changes in gene expression were shown to be transmitted to subsequent generations through aspartame-exposed males [122]. While this may be an example of paternal epigenetic inheritance, it is possible that mobile genetic elements expressed within the microbiome of the mice were

passed down to future generations. Regrettably, no data were presented on the microbiome of the strain of mice used in these experiments [122]. As we have previously reported, we consider anxiety to be the consequence of brain growth throughout childhood in the absence of a fully functioning gut–brain axis [27]. In general, however, it seems that great caution is needed for the interpretation of animal experiments.

Along the same lines, a study by Reese et al. on animal domestication found a parallel with human industrialisation in that certain bacterial species had apparently been lost [123]. Interestingly, the assigned reason followed the logic stated by Sonnenburg et al., being due to the more specialised diet of animals kept in captivity [101], but without looking for any evidence that such lower diversity animals may suffer from the atopic diseases described in pet and farm animals [48]. Whatever the ultimate reason for the observed loss of diversity, their conclusions were clearly correct; they noted the limitations of domesticated animal models and: "the importance of studying wild animals and non-industrialized humans for interrogating signals of host–microbial coevolution" [123].

A key point is that a wild animal population must represent each individual holobiont: a combination of host and fully functioning microbiome. Interestingly, mice are among the animals practising coprophagy, which presumably has the effect of homogenising their microbiomes and, in laboratory terms, this can have an influence [124]. It is also important to note that while the nature of the microbes may vary, for example, with blood group [125,126], it may well be the function that they bring to the microbiome that is more important—along with the flexibility of horizontal gene transfer.

### 13. The Search for a Standard Microbiome: Birth Microbes

While a large number of factors remain to be quantified, it seems clear that non-communicable disease including paediatric FGID is related to the loss of microbiome function, probably at an early "handshaking" phase in the infant. Accordingly, it is important to know which are the key features of an effective microbiome within the context of its transfer to a newborn. Indeed, the significance of horizontal gene transfer is that functions useful to the microbiome may be expressed by any of several compatible bacteria. Experience gained with the genes conferring antibiotic resistance to bacteria suggests that this compatibility can be very wide indeed [127,128]. As an example, one potential candidate, *Akkermansia muciniphila*, is considered to be the "gatekeeper" of the intestinal mucosa [129]. However, things are not always clear, and while bifidobacteria are generally considered to be "good bacteria", albeit with little actual evidence [130], it is interesting to note that they were not part of the constitution of the microbiome of the supposedly healthy Tanzanian Hadza people [95]. While the bulk of microbiome investigation has been carried out on bacteria, it is possible, bearing in mind the caveats of Brüssow, that any conclusions are erroneous [22]. Indeed, as stated above, it may be that this focus on bacteria is completely wide of the mark, and that microeukaryotes are actually the "missing link" between prokaryotes and the multicellular eukaryotes [43]. Key microbes are most likely to be found in "pre-industrial" societies such as those studied by Burkitt [19].

While the number of people living in unpolluted environments is rapidly decreasing, the fact that weight gain on entering a "Westernised" environment is still an ongoing process [131] implies that a substantial portion of some populations may retain a largely intact microbiome, at least in parts of the Venezuelan Amazon. As above-mentioned, bacterial microbiome-related work has been undertaken on the Tanzanian Hadza [95], along with studies on "uncontacted" Amerindians [132]. Although to our knowledge there are no published microbiome data as yet, the Bolivian Tsimane are fundamentally healthy but with a raised inflammatory level [133,134]. It is clear that a "standard", fully functioning microbiome will still exist somewhere, and of course, any profit gained from the exploitation of this fact must be shared with the community that provided it, according to suggestions related to anti-cancer work [135].

Once the evolutionary nature of the microbiome is accepted, so is the transfer of species-specific microbial entities mainly from the mother to child. While attention is

normally focussed on semiochemical-producing prokaryotes [40,54], microbial eukaryotes may also play a part, potentially within the immune system, for example, for fungi with respect to type 1 diabetes in children [65]. Importantly, as they may well be missing from an industrialised environment, it will be necessary to investigate microbes from both intestinal and intrauterine milieu within the context of the birth process [136] within true wild-type environments [123]. Accordingly, we can only be sure of the relevant constituents of the nascent microbiome if we can physically sample what is being transferred in this process on the body of the developing offspring or egg. These microbe-transfer tests can be carried out relatively easily on non-domesticated animals or non-industrialised humans, the only essential feature being the lack of Burkitt-type non-communicable disease throughout the population. While undoubtably intrusive, these tests have the benefit of being essentially non-invasive.

## 14. The Search for Semiochemical Function: An Ingestible Sensor

In order to interact with the gut–brain axis, some component of the gut wall, possibly the newly identified neuropod cells discussed earlier [76], has to recognise characteristic semiochemicals that may include dopamine or similar biogenic amines [77]. The analysis of faecal samples in the presence or absence of known microbiome-stimulating foods [31] may give an indication of the nature of these agents, while in vitro experiments on the isolated gut wall may succeed in identifying a complementary receptor. Disease-inducing microbiome-function deficiency may be due to the inadequate formation of such semio-chemicals, either directly in the adult or in the initial neonate handshaking process. If the latter, an "uncalibrated" gut wall may never be able to respond correctly as an adult. Regardless of the exact problem, an ingestible sensor calibrated to the detection of such semiochemicals is likely to prove invaluable in the solution of microbiome-related non-communicable disease [137].

The miniaturisation of electronics has allowed for the development of ingestible sensors: pill-like devices containing both a sensor and a transmitter capable of providing real-time information in a wide variety of health applications. Perhaps their most well-known use is in wireless capsule endoscopy, with lights and a camera [138]. However, they are now being assessed for an increasing array of applications for which they are classified as "minimally invasive" [139,140]. In principle, any robust detector system can be employed, provided that it is capable of being adequately miniaturised. One notable example is to use bacteria themselves as the detection agent, which could be useful with respect to the microbiome [141]. Interestingly, suggestions have made about the remote monitoring of vital signs, specifically an acoustic system to measure heart rate, in order to monitor athletes or service personnel operating in stressful or dangerous environments [142]. Bearing in mind the comments of Reese et al. on the "the importance of studying wild animals and non-industrialised humans for interrogating signals of host–microbial coevolution" [123], measuring semiochemical production as a function of everyday life, rather than being confined to sedentary laboratory studies, may offer more pertinent information [137]. Having obtained an indication of likely substances potentially including dopamine, the next step would be the construction of an ingestible sensor calibrated to their detection and subsequent use in human volunteers, initially from populations suffering from various levels of non-communicable disease. Subsequent work in populations not suffering from such diseases should afford complementary data relating to a "standard microbiome" (Section 13), while similar observations on wild-type animals, for example, pigs and primates, should be able to provide a degree of evolutionary context.

Once semiochemical behaviour is fully understood, possibly in response to a food challenge, it should be possible to assess the likelihood of a mother giving birth to a child with good prospects of future health, bearing in mind the fact that microbes, or at least bacteria, may be transferred to the neonate from areas other than the intestine [70]. A generic scheme linking such a challenge with ingestible sensor output to assess maternal microbial effectiveness is presented in Figure 4. However, judging by the prevalence of

non-communicable disease in the Burkitt-like "modern Western civilisation", it is likely that the majority of potential mothers would fail this ingestible sensor test.

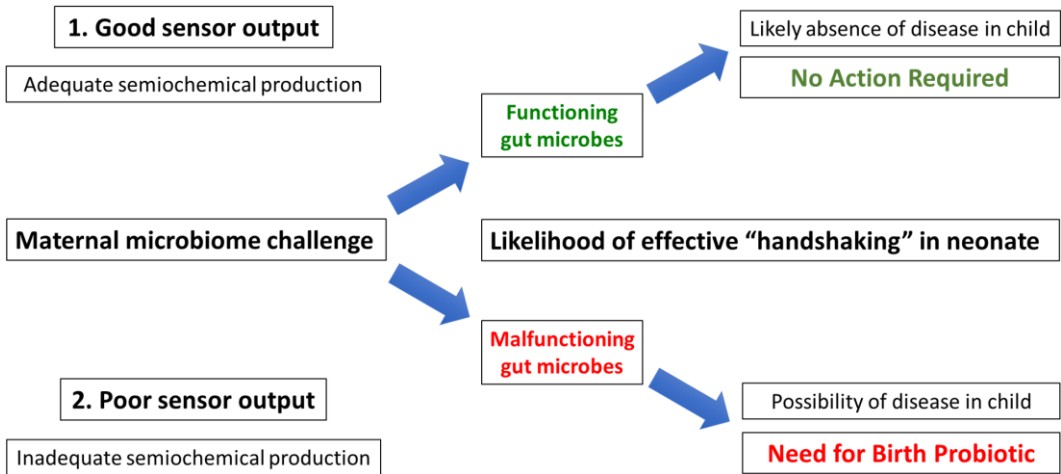

**Figure 4.** Generic use of ingestible sensor to assess maternal microbial effectiveness. In principle, an ingestible sensor can be used to check the microbiome of a woman preparing for childbirth. Left hand side: Represents the application of a food challenge to the microbiome of the prospective mother in the presence of an ingestible sensor calibrated to an appropriate semiochemical—the response of the sensor indicates the magnitude of semiochemical production. Right hand side: Assuming that the gut microbes are still working well at the time of birth, a good sensor response after this test should lead to a baby free from the non-communicable diseases described by Burkitt [19]. In contrast, a poor sensor response would point to the need for a birth probiotic to be given to the neonate, accounting for its genetic background.

Note that regardless of the "success" or "failure" of this investigation, it is imperative that the parents carry no moral blame if disease in the infant is considered to derive from the failure of the mother's microbiome.

## 15. Future Prevention of Paediatric FGID: A Birth Probiotic

Once the exact role of key microbes have been identified, it should be possible to assemble a "birth probiotic" potentially capable of restoring the industrialised human microbiome back to health, and to freedom from the non-communicable diseases documented by Denis Burkitt [19]. As there is little evidence for the incorporation of probiotic microbes into adult microbiomes [130], it will probably be necessary to add such birth microbes to the baby after the fashion of vaginal inoculation [62]. As described above, in principle, the addition of such "birth microbes" to the neonate would allow for the return of at least some function, although, bearing in mind the possible need for sentinel cells to experience prior exposure to local microbes, the full calibration of the neonate immune system may require a further generation. However, a significant improvement in population health may quickly become apparent.

Needless to say, going back to the levels of malnutrition and infectious disease commonly seen in ages past should be avoided. Equally, we must treat heavy metal toxins with the care that they deserve. Accordingly, the following precepts should be followed.

### 15.1. Food

Our previous studies involved the amelioration of non-communicable disease with the aid of dietary fibre and polyphenols [31], along with the role of probiotics in stimulating the immune/semiochemical system [27]. In addition, pregnancy requires greater respect for behavioural [143] and dietary [144] constraints.

*15.2. Antibiotics*

In essence, antibiotics work most effectively in conjunction with fully functional immune and lymphatic systems. While the antibiotic disables the pathogen, preventing its growth, it allows the immune system to out-compete it. In turn, the lymphatic system drains away the debris, thereby preventing the formation of a viral shunt [98]. In principle, the virtue of so-called phage therapy is that the viral agent can be more precisely targeted at the pathogen [145], thus minimising damage to the microbiome. However, it could be that injected antibiotics will prove to be both more effective than oral antibiotics as well as less prone to causing resistance.

*15.3. Heavy Metal Toxins*

Although the overt toxicity of lead and other heavy metal poisons has long been recognised [113], microbiome-function deficiency disease is covert and tends to affect the following generations more severely [66]. The use of lead, for example, increases in more economically active periods, so the rise and fall of empires have been tracked in Greenland ice [146]. Of course, poison operates at an individual level, and just as Bostock noted in his hay fever [112], we have previously suggested that the so-called "Venus" figurines from the Palaeolithic [147] could be due to microbiome poisoning among the leading families of the day [27]. The modern use of precious metal catalysts may leave the same problems for the future [148].

**Author Contributions:** D.S. and S.J.: Concept design, hypothesis consideration, and original draft preparation; M.P.-P. and G.I.L.-C.: Manuscript draft and related research; D.S., M.P.-P. and G.I.L.-C.: Proofreading and suggestions. H.V.F. and B.S.: Additional references. All authors have read and agreed to the published version of the manuscript.

**Funding:** M.P.-P. is a postdoctoral researcher fellow with CVU 694877 from Consejo Nacional de Humanidades, Ciencias y Tecnologías (CONAHCYT) at the Universidad Nacional Autónoma de México (UNAM). G.I.L.-C. is a postdoctoral researcher fellow from DGAPA-PAPIIT IV201220, UNAM.

**Institutional Review Board Statement:** Not applicable.

**Informed Consent Statement:** Not applicable.

**Data Availability Statement:** Not applicable.

**Acknowledgments:** We thank Mariana Colmenares Castaño, Leslie Elizabeth Félix Oropeza, and María Fernanda Cardona Herrera for their helpful discussions. We also thank the anonymous reviewers.

**Conflicts of Interest:** The authors declare no conflict of interest.

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
