# Peer review of "On the Inheritance of Microbiome-Deficiency: Paediatric Functional Gastrointestinal Disorders, the Immune System and the Gut–Brain Axis"

_gastrointestdisord, doi:10.3390/gidisord5020018_

Round 1

Reviewer 1 Report

David Smith and colleagues wrote their review manuscript “ On the Inheritance of Microbiome-Deficiency: Paediatric Functional Gastrointestinal Disorders, the Immune System and the Gut-Brain Axis” in an informative and interesting way.

I liked that the authors gave their own opinion throughout the text.

I have a few minor corrections to propose:

Line 16 – delete “that have”

Lines 23-25. The authors wrote: “Our tentative conclusion is that the fully functioning, mutualistic microbiome has a single primary role: to transfer antigen information from the mother to the neonate in order to calibrate its immune system, allowing it to survive within the microbial environment into which it will emerge.”. What about males? Is their microbiome also mutualistic but without its primary role?  

Line 32: inherits… the microbiome

Line 94: space after 1911

Iine 169: I wouldn’t write “electronic”.  I would prefer informatics or computer science

Lin 277-278. Please rephrase “every healthcare professional has a different opinion.”.

Line 289-290. Please rephrase “but, of course, they did not check for the presence of unicellular eukaryotes.”. For example  delete “of course”

292: I did not understand this sentence. Was that study performed with just one child?

Line 295: “can only worsen across the generations”. Why? Can’t they receive those microbes during this life?

Line 299 Please rephrase “for whatever reason”

Lines 512-513 I think the authors should rephrase lines 512-513. For example, “ In 1928, Frederick Griffith published some observations on the transfer of lethality between two such strains of S. pneumoniae”.

Lines 559 – 572: its sentence should be rephrased

Line 616: delete “implies”. I thing it should be “does not imply”

Line 622: delete the “ : ”

Line 693: What happens with the sweetener aspartame? Could you elaborate more on this?

I explained all the corrections the author should perform in my list of minor corrections.

Author Response

We thank the Reviewer 1 for the valuable comments. Please see attachment for a breakdown of our comments.

Reviewer 2 Report

The manuscript:  On the Inheritance of Microbiome-Deficiency: Paediatric Functional Gastrointestinal Disorders, the Immune System and the Gut-Brain Axis

David Smith, Sohan Jheeta, Georgina I. López-Cortés, Bernadette Street, Hannya V. Fuentes, Miryam Palacios-Pérez

Title: corresponds to the content of the article

Introduction: enough; represent the essence of the problem; does not require change

Methodology: meets the requirements of the journal and the branch of knowledge; does not require modification;

Statistics: meets the requirements of international standards

Results: clearly stated and do not require changes

Discussion of the results: sufficient and consistent with the main results obtained.

Literature: sufficient for the article and does not require processing

Article design: meets the requirements of the journal

Conclusion: the article meets the requirements of the journal and can be published without significant revision

Author Response

We are honoured that our work has been appreciated by Reviewer 2.